# DDK: Distilling Domain Knowledge for Efficient Large Language Models

**Jiaheng Liu*[†,1], Chenchen Zhang*[2], Jinyang Guo[3], Yuanxing Zhang[2], Haoran Que[1],**
**Ken Deng[1], Zhiqi Bai[1], Jie Liu[4], Ge Zhang[5], Jiakai Wang[2], Yanan Wu[1], Congnan Liu[1],**
**Jiamang Wang[2], Lin Qu[2], Wenbo Su[1], Bo Zheng[1]**
[1]Taobao & Tmall Group of Alibaba, [2]Alibaba Group, [3]The University of Sydney,
[4]The Chinese University of Hong Kong, [5]University of Waterloo
{ljh411989}@alibaba-inc.com

## Abstract

Despite the advanced intelligence abilities of large language models (LLMs) in various applications, they still face significant computational and storage demands. Knowledge Distillation (KD) has emerged as an effective strategy to improve the performance of a smaller LLM (i.e., the student model) by transferring knowledge from a high-performing LLM (i.e., the teacher model). Prevailing techniques in LLM distillation typically use a black-box model API to generate high-quality pretrained and aligned datasets, or utilize white-box distillation by altering the loss function to better transfer knowledge from the teacher LLM. However, these methods ignore the knowledge differences between the student and teacher LLMs across domains. This results in excessive focus on domains with minimal performance gaps and insufficient attention to domains with large gaps, reducing overall performance. In this paper, we introduce a new LLM distillation framework called DDK, which dynamically adjusts the composition of the distillation dataset in a smooth manner according to the domain performance differences between the teacher and student models, making the distillation process more stable and effective. Extensive evaluations show that DDK significantly improves the performance of student models, outperforming both continuously pretrained baselines and existing knowledge distillation methods by a large margin.

## 1 Introduction

Recent advancements in Large Language Models (LLMs) such as LLaMA [7, 9, 58, 59] have garnered significant attention due to their strong intelligence. However, these models also impose considerable computational and storage demands, particularly in practical deployments such as instant chat, copilot, and query rewriting. Consequently, the development of lightweight yet efficacious LLMs suitable for real-world applications has become an area of increasing research interest. Several small-scale LLMs, e.g., Phi [39] and MiniCPM [32], have been designed to facilitate rapid inference on devices with limited resources. These models are generally trained from scratch using a large volume of selectively curated high-quality datasets, which could be prohibitive for the broader research community. Meanwhile, there has been a surge in the exploration of model compression techniques [40] to reduce the resource footprint of LLMs. Apart from these techniques, knowledge distillation (KD) emerges as a prominent method for creating effective neural networks, which transfer knowledge from a high-performing teacher model to a compact student model.

---

* First two authors contributed equally.
[†] Corresponding Author: Jiaheng Liu.

38th Conference on Neural Information Processing Systems (NeurIPS 2024).

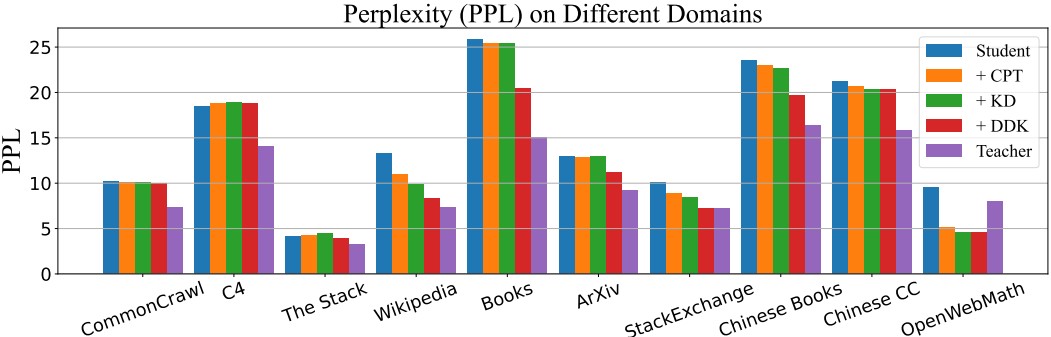

Figure 1: The perplexity scores of different methods across different domains for different methods (See Section 4 for more details.). Note that "Chinese CC" denotes "Chinese CommonCrawl".

The primary challenges in enhancing the performance of KD approaches on LLMs stem from two main aspects: i) appropriately utilizing the data [3, 67]; ii) stabilize the distillation process [68]. Recently, it has become increasingly acknowledged that the mixture ratios of various domains within the training dataset substantially affect the performance [20, 67, 69]. Regarding the issue of data composition, the influence of domain-specific mixtures for KD remains underexplored. As shown in Fig. 1, the performance between Qwen-1.5 1.8B [7] (student) and Qwen-1.5 14B [7] (teacher) reveals that the performance gap varies significantly across domains. For instance, in the "Books" domain, the student model significantly underperforms the teacher model, while in "The Stack" domain, the difference is minimal, which indicates that the "Books" domain is relatively not optimized well for the student model compared to the teacher model, and more data from the "Books" domain should be included. Therefore, we aim to design a knowledge distillation framework that can dynamically adjust the data composition during distillation to reallocate more computation to domains, where the student and teacher models have larger performance gaps.

In this paper, we introduce a novel methodology, termed **D**istill **D**omain **K**nowledge for LLMs (DDK), which effectively optimizes domain-specific mixtures to address the performance discrepancy between teacher and student models across different domains. Specifically, DDK begins by quantifying the performance deviations between the teacher and student LLMs using an offline-collected validation dataset across various domains. Next, it periodically re-calculates the domain discrepancy factor based on the performance gap between the teacher and student models. Finally, DDK employs a domain knowledge-guided sampling strategy to sample data from different domains with varying probabilities based on the calculated domain discrepancy factor. Additionally, inspired by the optimization algorithms [36], we propose a factor smooth updating mechanism to augment the stability and robustness of the DDK approach. For the supervision loss, we minimize the differences in the output logits between the teacher and student models. As demonstrated in Fig. 1, the performance gap across domains is significantly reduced by DDK.

Our main contributions are summarized as follows:

- To the best of our knowledge, we are the first to study the influence of domain-specific data mixtures for distilling LLMs, and efficiently transfer the domain knowledge of the teacher network upon the domain weights.
- DDK proposes a factor smooth updating strategy to strategically enhance the appropriate focus of the distillation process on targeted domains, which effectively stabilizes the domain knowledge guided sampling process for smoother distillation.
- Extensive experiments on multiple benchmark datasets demonstrate the effectiveness and generalization ability of our proposed DDK.

## 2 Related Works

**Large Language Models.** The emergence of LLMs [62, 72, 24, 41, 19, 65, 49, 53, 27, 6, 66] marks a significant milestone in the domain of natural language processing, with notable examples including GPT3, Lamda, Palm, and several others [2, 4, 10, 44, 57]. For example, Radford and Narasimhan

[50] introduced the GPT model, leveraging multiple layers of transformer decoder blocks, while Meta later developed LLaMA [58] employing an enhanced transformer architecture, subsequently evolved into LLaMA2 [59]. Recent advancements have also seen the application of instruction tuning [13, 63] and learning through human feedback [8, 45, 74] to better align LLMs with human understanding and foster the creation of versatile AI assistants [21, 43]. Despite their potential, LLMs' extensive capabilities are often accompanied by vast sizes [35, 64], demanding significant computational resources. In this work, we aim to focus on how to produce small LLMs based on the knowledge distillation approach.

**Knowledge Distillation**. Knowledge distillation is a pivotal technique in model compression and acceleration [61, 42, 26, 70, 25], primarily employed to transfer knowledge from a robust, well-trained teacher model to a compact student model [29, 1, 37]. Recently, several approaches to knowledge distillation tailored for LLMs have been proposed. These approaches can be broadly classified into two categories: *White-box KD* leverages either the internal parameters or the logits of the teacher LLM during the distillation process [22, 46, 56, 71]. For example, Gu et al. [23] propose that traditional Kullback-Leibler divergence (KLD) objective is inappropriate for open text generation tasks and propose MiniLLM to minimize reverse KLD through policy gradient techniques [54]. Conversely, *black-box KD* relies solely on the outputs from the teacher model [12, 30, 34, 48, 60]. For example, "Distilling Step-by-Step" strategy [31] employs Chain of Thought (CoT) prompting to provide sophisticated guidance during distillation. These two types of KD approaches mainly focus on aligning the generative behaviors of the teacher and student models. DDK delves into the efficacies of domain-specific distillation, aiming to mitigate the discrepancies in performance between the teacher and student model across different domains. Hence, DDK is fundamentally orthogonal to these methods.

# 3 Methodology

## 3.1 Overview

Figure 2 illustrates the comprehensive architecture of the DDK framework. DDK employs a large-scale teacher LLM and a comparatively smaller student LLM, with the objective of transferring knowledge from the former to the latter to enhance performance utilizing a specially curated distillation dataset. Initially, the distillation dataset is constructed by randomly sampling from the training corpus. Throughout the distillation process, we continuously assess the domain-specific performance of both the teacher and student LLMs, and use domain knowledge guided sampling to dynamically update the data mixture on the student's abilities within specific domains. As the domain proficiency of the student LLM evolves during distillation, we introduce a factor smooth updating strategy to ensure the robustness of the domain knowledge-guided sampling approach. Finally, DDK provides of a better student LLM, optimized for enhanced performance across targeted domains.

## 3.2 Domain Knowledge Guided Sampling

The distilled student LLMs are anticipated to exhibit robust competence across various preset domains. Nevertheless, prevailing knowledge distillation techniques tailored for LLMs tend to homogeneously optimize performance across these domains, leading to potential performance degradation. To address this issue, we design the domain knowledge guided sampling strategy to enhance distillation efficacy by prioritizing domain-specific complexities.

**Domain discrepancy factor construction.** We consider a dataset $\mathcal{D}$ that has been partitioned into $N$ distinct domains. We denote the pre-trained teacher LLM as $\mathcal{M}_T$ and the student model, which is currently under training, as $\mathcal{M}_S$. To efficiently identify and prioritize data that may yield the most learning benefit, particularly from domains where the student model underperforms, we introduce a *domain discrepancy factor* denoted as $\mathbf{r} \in \mathbb{R}^N$. Each component $\mathbf{r}[i]$ of this vector quantitatively represents the discrepancy in performance between the teacher and student models within the $i$-th domain. As we assume a good student should exhibit close approximation to the teacher across all domains, $\mathbf{r}$ is calibrated to reflect differential performance indices as follows:

$$\mathbf{r}[i] = \exp(\ell_S[i]/\ell_T[i])/ \sum_{i' \in \{1,...,N\}} \exp(\ell_S[i']/\ell_T[i']) \tag{1}$$

where $\ell_S[i] = \exp(\mathrm{CE}(\mathcal{M}_S(V_i), Y_i))$ and $\ell_T[i] = \exp(\mathrm{CE}(\mathcal{M}_T(V_i), Y_i))$.

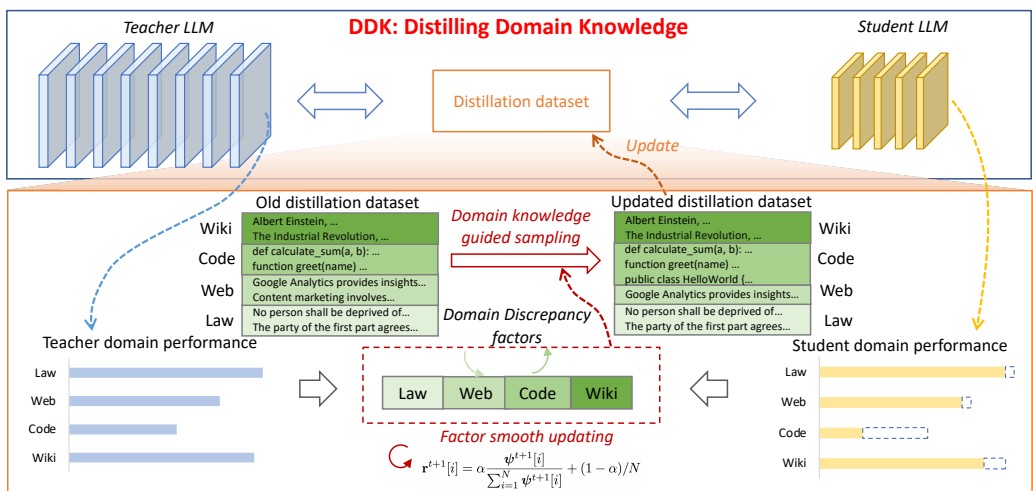

Figure 2: Overview of the distillation process of DDK. First, the training dataset is divided into distinct domains based on predefined criteria. Then, DDK dynamically modulates the distribution of domain-specific data, augmenting the amount allocated to domains where the student model struggles the most. The proportions attributed to each domain are recalculated at distillation intervals by employing a factor smooth updating approach.

Here, $V_i$ and $Y_i$ are the inputs and the ground-truth labels of the validation dataset of the $i$th domain. $CE(\cdot)$ represents the cross-entropy loss. $\ell_S \in \mathbb{R}^N$ and $\ell_T \in \mathbb{R}^N$ are the perplexity scores over the validation sets of all domains for student and teacher respectively, indexed by the domain index $i$. In this case, a higher value of $\mathbf{r}[i]$ signifies a pronounced disparity in domain-specific proficiency between the student model and the teacher model. Accordingly, it is imperative to allocate more relevant data to enhance the domain expertise.

**Domain knowledge guided sampling.** We employ a domain knowledge-informed sampling strategy to refine the composition of the distillation dataset, which utilizes a probabilistic mechanism defined by vector $\mathbf{r}$ to iteratively select samples from the training corpus. The process continues cyclically once a domain data has been exhausted. Finally, DDK strategically increases the data allocation towards underperforming domains, thereby mitigating the performance discrepancies between the teacher and student models across all domains.

### 3.3 Factor Smooth Updating

With the domain knowledge guided sampling strategy, we can dynamically focus on more challenging domains during the distillation process. Nonetheless, we observe that the domain discrepancy factor exhibits significant fluctuations throughout this procedure. Such rapid alterations may precipitate exceedingly unbalanced data sampling, potentially compromising the stability of the distillation.

**Factor smooth updating.** To enhance the stability of the distillation process, we periodically adjust the domain discrepancy factor every $K$ iterations throughout the distillation process, thereby partitioning it into discrete intervals. The parameter $K$ is pivotal as it governs the system's capacity to address immediate discrepancies and influences the stability of the data mixture. We denote the domain discrepancy factor for the $i$-th domain at the $t$-th interval of distillation as $\mathbf{r}^t[i]$. Similarly, let $\ell_S^t[i]$ and $\ell_T^t[i]$ denote the perplexity scores at the beginning of the $t$-th distillation interval. In DDK, the domain discrepancy factor at the $(t+1)$-th interval is defined as:

$$\mathbf{r}^{t+1}[i] = \alpha \frac{\boldsymbol{\psi}^{t+1}[i]}{\sum_{i=1}^{N} \boldsymbol{\psi}^{t+1}[i]} + (1-\alpha)/N,$$

$$\text{where } \boldsymbol{\psi}^{t+1}[i] = \mathbf{r}^t[i]\exp(\ell_S^{t+1}[i]/\ell_T^{t+1}[i]). \tag{2}$$

Note that a constant term is incorporated in $\mathbf{r}^t[i]$ to preclude the occurrence of excessively small values, thereby guaranteeing a baseline probability for data sampling across various domains. The parameter $\alpha$, designated as the smoothing coefficient, is fixed at a value of 0.5 in our experimental

---

**Algorithm 1** Distillation procedure of the DDK framework.

---

**Input:** Distillation dataset $D$; The steps per distillation interval $K$;
 1: Initialize domain discrepancy factor $\mathbf{r}^0$ based on Eq. 1;
 2: Randomly sample $D^0 \subset D$ that supports $K$ steps distillation;
 3: Initialize student training iteration $c = 0$, distillation interval $t = 0$;
 4: **for** each iteration in the training process **do**
 5:     // Update student LLM parameters
 6:     Read a batch of samples and use Eq. 3 to update the parameters of student LLM;
 7:     $c = c + 1$
 8:     **if** $c \bmod K == 0$ **then**
 9:         // Update distillation data mixture
10:         $t = t + 1$;
11:         Use Eq. 2 to update domain discrepancy factor $\mathbf{r}^t$;
12:         Sample a dataset, $D^t \subset D$, that supports $K$ steps distillation according to $\mathbf{r}^t$;
13:         Shuffle $D^t$;
14:         **if** $t$ reaches a preset maximal number of intervals **then**
15:             Stop the distillation loop;
**Output:** The distilled student LLM;

---

setup. In addition, the inclusion of $\psi^t$ imparts a history mixture information on the modification of the domain discrepancy factor. This mechanism facilitates a gradual modification of $\mathbf{r}^t[i]$, thereby minimizing fluctuations and ensuring a stable, domain knowledge-driven distillation process for fetching informative data.

## 3.4 Overall Optimization

As we jointly update the student LLM parameters and the domain discrepancy factor in the distillation process, the optimization object can be written as follows:

$$\min_{\theta_S} \sum_{i \in \{1,\dots,N\}} \text{CE}(\mathcal{M}_S(V_i), Y_i) + \gamma \text{KL}(\text{Softmax}(z_S(V_i), T), \text{Softmax}(z_T(V_i), T)), \tag{3}$$

where $\theta_S$ is the parameters of the student model. $z_S(\cdot)$ and $z_T(\cdot)$ are the output hidden states from student and teacher LLMs, respectively. We leverage KL-divergence to approximate the student model's output to the teacher model's output, over a distillation temperature $T$. $\gamma$ is the factor to balance these two terms. Algorithm 1 summarizes the pseudo-code of the DDK process. In practice, the distillation process is typically concluded either when all available data has been fully utilized or when the domain discrepancy factor approaches a threshold indicative of minimal disparity between the teacher and student models.

# 4 Experiments

In this section, we make comprehensive evaluations to answer two research questions: **RQ1**: To what extent does the DDK process improve the performance of a small-scale LLM? **RQ2**: How does the dynamic domain-specific guidance contribute to the overall improvement?

## 4.1 Experimental Setup

**Model configuration details.**  We use the Qwen-1.5 [7] and LLaMA2 [59] to demonstrate the effectiveness of DDK. Regarding the Qwen-1.5 series, we use Qwen-1.5 14B and Qwen-1.5 1.8B as the teacher and student models, respectively. For LLaMA2 series, we use LLaMA2 13B and TinyLLaMA 1.1B [73] as the teacher and student models, respectively.

**Training details.**  Due to the unavailability of training data for LLaMA2 and Qwen-1.5 models, we mainly utilize RedPajama [16] for distillation, which consists of training data derived from seven distinct domains: CommonCrawl, C4, The Stack, Wikipedia, Books, ArXiv, and StackExchange.

Table 1: Results of different methods on the Qwen-1.5 models. Note that we use Qwen-1.5 14B and Qwen-1.5 1.8B as teacher and student models, respectively. "W.G.", "C.QA" and "H.E." denote Winogrande, CommonsenseQA and Humeneval datasets, respectively.

| Methods | CEval | MMLU | RACE | C3 | W.G. | GSM8K | C.QA | Arc-E | Arc-C | H.E. | MBPP | Avg. |
|---|---|---|---|---|---|---|---|---|---|---|---|---|
| Teacher (14B) | 78.68 | 64.34 | 89.95 | 77.38 | 68.74 | 67.63 | 82.06 | 87.58 | 80.59 | 37.80 | 44.00 | 70.80 |
| Student (1.8B) | 59.66 | 44.48 | 69.57 | 58.27 | 57.85 | 38.4 | 64.70 | 70.23 | 50.31 | 11.87 | 18.00 | 49.39 |
| + CPT | 60.13 | 45.01 | 69.00 | 60.30 | 56.98 | 42.50 | 64.78 | 72.00 | 51.03 | 13.12 | 20.45 | 50.48 |
| + KD [29] | 61.29 | 43.63 | 70.12 | 63.92 | 58.01 | 49.58 | 66.26 | 73.41 | 54.56 | 15.63 | 25.15 | 52.87 |
| + TED [40] | 62.04 | 45.21 | 69.95 | 63.18 | 57.38 | 49.28 | 65.27 | 74.74 | 55.00 | 13.75 | 22.69 | 52.59 |
| + MiniLLM [23] | 61.66 | 45.07 | 68.92 | 63.37 | 57.14 | 48.90 | 64.46 | 74.52 | 53.92 | 16.88 | 23.55 | 52.58 |
| **+ DDK (Ours)** | 63.75 | 46.01 | 71.56 | 65.53 | 59.10 | 53.54 | 66.75 | 75.01 | 55.03 | 27.13 | 26.10 | **55.41** |

Table 2: Results of different methods on the LLaMA models. Note that we use LLaMA2 13B and TinyLLaMA 1.1B as teacher and student models, respectively.

| Methods | CEval | MMLU | RACE | C3 | W.G. | GSM8K | COSE-QA | Arc-E | Arc-C | H.E. | MBPP | Avg. |
|---|---|---|---|---|---|---|---|---|---|---|---|---|
| Teacher (13B) | 34.32 | 49.31 | 62.85 | 46.03 | 63.77 | 24.10 | 52.17 | 73.30 | 49.40 | 18.30 | 28.10 | 45.60 |
| Student (1.1B) | 23.92 | 24.89 | 22.92 | 35.24 | 55.49 | 14.19 | 19.08 | 24.18 | 24.12 | 5.62 | 16.58 | 24.20 |
| + CPT | 26.79 | 26.26 | 24.24 | 38.91 | 56.20 | 15.03 | 20.39 | 28.06 | 26.03 | 6.88 | 17.35 | 26.01 |
| + KD | 27.12 | 26.13 | 23.84 | 37.43 | 53.91 | 15.92 | 22.52 | 29.40 | 26.27 | 7.50 | 17.97 | 26.18 |
| + TED | 27.49 | 26.43 | 24.18 | 37.61 | 55.72 | 14.74 | 22.93 | 28.61 | 25.40 | 8.13 | 17.45 | 26.24 |
| + MiniLLM | 26.74 | 26.45 | 24.32 | 37.18 | 54.46 | 16.30 | 22.93 | 29.46 | 25.84 | 8.13 | 18.28 | 26.37 |
| **+ DDK (Ours)** | 27.86 | 28.74 | 27.76 | 42.41 | 57.62 | 17.44 | 25.39 | 36.29 | 30.15 | 9.36 | 19.51 | **29.32** |

Moreover, to enhance the model's proficiency in Chinese and Mathematics, we also incorporate three cleaned open-source datasets (i.e., Chinese Books [19], Chinese CommonCrawl [19], and OpenWebMath [47]). Therefore, there are ten domain datasets for the distillation. In addition, to assess the disparity in performance between teacher and student models across the ten domains, we have constructed a domain-specific validation set for each domain, where each domain includes 500 samples. During the distillation phase, the student models are trained on approximately 15B tokens. For the training framework, we employ the DeepSpeed-Chat code[1] as our codebase, and conduct all experiments using 16 NVIDIA A100 GPUs (80G), where FlashAttention V2 [17] is used to accelerate training. For the training schedule, we first apply the warm-up strategy to increase the learning rate from 0 to $3e^{-5}$ in 1,000 steps. Then, we use the cosine learning rate schedule, where the final learning rate is $3e^{-6}$ and the whole training step is about 30,000 steps. Empirically, we set the distillation interval $K$ as 1,000 and the temperature $T$ as 1.0.

**Evaluation details.** As we do not conduct instruction tuning on the student models, we mainly report the zero-shot, close-ended results across commonly used datasets including C-Eval [33] (val), MMLU [28] (test), RACE [38] (high, test), C3 [52] (test), WinoGrande [51] (val), GSM8K [15] (test), CommonsenseQA [55] (val), Arc-E [14] (test), Arc-C [14] (test) and HumanEval [11] (test). We also report the 3-shot performance on MBPP [5] (test).

**Baseline details.** We compare DDK with four baseline methods:

- **CPT** denotes that we continue to pre-train the student model by using the same number of training tokens without considering domains.

- **KD** [29] denotes the standard knowledge distillation by computing the KLD between the teacher and student logits without considering domains.

- **TED** [40] denotes to use task-aware filters to align the hidden representations of the student and the teacher at each transformer layer.

- **MiniLLM** [23] denotes to replace the forward KL divergence with reverse KL divergence, which prevents the student model from overestimating the low-probability regions of the teacher distribution.

---

[1] https://github.com/microsoft/DeepSpeedExamples/tree/master/applications/DeepSpeed-Chat

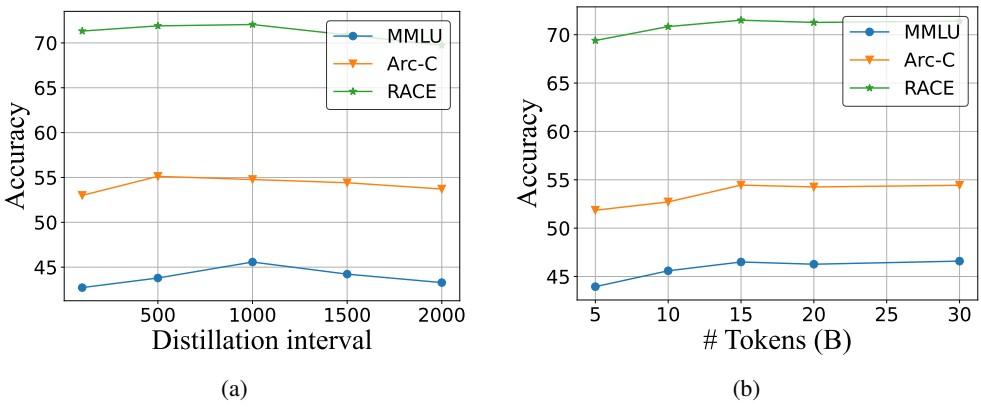

(a)

(b)

Figure 3: (a). Effect of distillation interval. (b). Effect of the number of training tokens.

## 4.2 Main Results

As shown in Table 1-2, we report the performance results of different baseline methods. The following observations provide a comprehensive response to RQ1: (1) We see that the absence of knowledge transfer from the teacher model significantly impedes the student model's capabilities in intricate tasks such as coding (e.g., HumanEval) and Chinese comprehension (e.g., C3). (2) DDK outperforms other baseline methods when using different types of teacher and student models, which demonstrates the effectiveness of DDK for training small student LLMs. (3) The baseline methods KD, TED, and MiniLLM exhibit similar performance. For instance, the average accuracy of these three approaches hovers around 52% when distilling onto the Qwen student model. We hypothesize that in the context of LLM distillation, domain data mixture may emerge as a key performance bottleneck, and the existing baseline techniques fail to adequately address this challenge. (4) The performance gains vary across different domains. Notably, when distilling the Qwen model, we achieve significant improvements on the reasoning tasks (e.g., Code on Humaneval and MBPP, Math on GSM8K), which indicates that the student model can improve a lot on the reasoning tasks under the guidance of the teacher model. This empirical observation suggests that DDK is successful in directing additional attention toward the more challenging problem domains.

## 4.3 Ablation Study

In this section, we perform ablation studies to assess the robustness of the DDK model and its sensitivity to key hyperparameters. We collected data using Qwen 1.5 and reported its performance on the validation sets of MMLU, RACE, and ARC-C, which differ from those discussed in the previous subsection. Initially, we concentrate on addressing RQ1 through fine-grained analyses.

**Effect of data sampling strategies.** We propose two variants of data sampling strategies on DDK. For DDK (w/o FS), we just remove the factor smooth updating mechanism and directly take $\mathbf{r}^t$ as the probability of each domain. For DDK (ES), we sample data from each domain equally. The results are shown in Fig. 4, and we can suppose that both factor smooth updating and domain knowledge guided sampling contribute to the distillation owing to the existence of domain-specific discrepancy.

**Effect of distillation interval.** Fig. 3 (a) shows the evaluation results on the effect of the distillation interval hyperparameter (i.e., $K$) in Alg. 1. We observe that increasing $K$ from 100 to 1,000

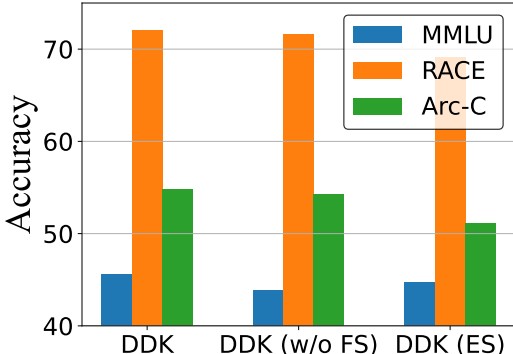

Figure 4: Effect of data sampling strategies.

leads to better performance, indicating that a rapid updating frequency may destabilize the distillation process. However, further increasing $K$ leads to inferior results. We conclude that when the updating

frequency is small, the domain weights update quickly and the student LLM weights can not be sufficiently optimized for the current distillation interval. Meanwhile, when the updating frequency is large, there is insufficient alignment between the LLM weights and the optimal domain weights.

**Effect of training tokens.** As shown in Fig. 3 (b) , we investigate the relationship between the results on three representative datasets and the number of training steps for Qwen-1.5 1.8B model when using Qwen-1.5 14B model as teacher. At the first 10B tokens, the results improve quickly, which indicates that the student models can benefit a lot with the supervision of the teacher model. When further increasing the training iterations, we observe that the performance tends to plateau, which indicates a fast convergence of distillation by DDK.

## 4.4 Further Analysis

We provide further investigation to show the applicability of DDK across more scenarios.

**Generalization ability of using different teacher / student models.** To show the generalization ability of DDK on different student models, we use Qwen-1.5 14B as the teacher model and use Qwen-1.5 4B as the student model. As shown in Table 3, DDK surpasses the baseline methods by a large margin. Additionally, comparative analysis with the enhancements observed when employing Qwen-1.5 1.8B as the student model, as presented in Table 1, verifies that a more capable student model tends to yield superior performance improvements. We then apply another teacher model to show the generalization ability of DDK. Specifically, we take Qwen-1.5 7B and Qwen-1.5 1.8B as teacher and student models, respectively. As documented in Table 4, DDK consistently facilitates the most substantial enhancement. These results demonstrate the efficacy and robustness of DDK in leveraging diverse teacher-student model configurations.

**Generalization ability on Code LLMs.** We implement DDK on LLMs, selecting the Code LLM StarCoder as a case study for empirical evaluation. Within the StarCoder series, we deploy StarCoder 15.5B as the teacher model and StarCoder 3B as the student model. The training corpus is primarily derived from four programming language domains—Python, Java, TypeScript, and C#—sampled from The Stack V2 dataset [2], with each language representing a distinct domain. We report the performance on the repository-level code completion dataset (i.e., CrossCodeEval [18]). The results in Table 5 affirm that DDK brings notable enhancements in the performance of StarCoder 3B, thereby highlighting DDK's efficacy in tackling the vertical distillation task.

Table 3: Results of different methods on the Qwen-1.5 models. Note that we use Qwen-1.5 14B and Qwen-1.5 4B as teacher and student models, respectively.

| Methods | CEval | MMLU | RACE | C3 | W.G. | GSM8K | C.QA | ARC-E | ARC-C | H.E. | MBPP | Avg. |
|---|---|---|---|---|---|---|---|---|---|---|---|---|
| Teacher (14B) | 78.68 | 64.34 | 89.95 | 77.38 | 68.74 | 67.63 | 82.06 | 87.58 | 80.59 | 37.80 | 44.00 | 70.80 |
| Student (4B) | 67.60 | 53.23 | 80.17 | 65.26 | 64.08 | 52.24 | 74.24 | 79.30 | 66.20 | 25.60 | 29.20 | 59.74 |
| + CPT | 68.05 | 52.78 | 79.56 | 67.72 | 63.61 | 54.00 | 74.32 | 80.20 | 66.67 | 26.30 | 31.00 | 60.38 |
| + KD | 68.35 | 52.90 | 80.13 | 70.31 | 63.53 | 56.00 | 75.51 | 82.19 | 67.18 | 27.50 | 32.85 | 61.50 |
| + MiniLLM | 68.20 | 51.93 | 79.22 | 68.78 | 62.27 | 55.72 | 73.87 | 83.92 | 67.37 | 28.13 | 33.05 | 61.13 |
| + DDK (Ours) | 68.57 | 53.17 | 82.53 | 70.25 | 64.85 | 62.09 | 75.14 | 84.10 | 68.95 | 30.63 | 39.12 | **63.58** |

Table 4: Results of different methods on the Qwen-1.5 models. Note that we use Qwen-1.5 7B and Qwen-1.5 1.8B as teacher and student models, respectively.

| Methods | CEval | MMLU | RACE | C3 | W.G. | GSM8K | C.QA | ARC-E | ARC-C | H.E. | MBPP | Avg. |
|---|---|---|---|---|---|---|---|---|---|---|---|---|
| Teacher (7B) | 74.10 | 58.39 | 85.78 | 76.03 | 65.59 | 54.53 | 79.28 | 85.78 | 72.30 | 35.63 | 37.40 | 65.89 |
| Student (1.8B) | 59.66 | 44.48 | 69.57 | 58.27 | 57.85 | 38.4 | 64.70 | 70.23 | 50.31 | 11.87 | 18.00 | 49.39 |
| + CPT | 60.13 | 45.01 | 69.00 | 60.30 | 56.98 | 42.50 | 64.78 | 72.00 | 51.03 | 13.12 | 20.45 | 50.48 |
| + KD | 62.63 | 45.07 | 69.86 | 61.18 | 57.08 | 48.14 | 65.27 | 73.74 | 52.50 | 13.75 | 22.69 | 51.99 |
| + MiniLLM | 62.40 | 45.20 | 69.10 | 61.45 | 57.46 | 47.56 | 65.11 | 73.86 | 52.97 | 14.38 | 23.31 | 52.07 |
| + DDK (Ours) | 64.41 | 46.44 | 70.98 | 63.37 | 57.54 | 54.06 | 66.83 | 74.43 | 55.09 | 11.88 | 24.98 | **53.64** |

**Analysis on the in-context learning abilities.** We evaluate in-context learning capabilities utilizing DDK and the other baselines through several few-shot benchmarks in Table 6. As shown in Table 6, we observe that our DDK consistently manifests considerable enhancements in performance, affirming that DDK transcends mere static knowledge transfer to the student model and augments the in-context learning capacity greatly.

---

[2] https://huggingface.co/datasets/bigcode/the-stack-v2

Table 5: Results of different methods on the StarCoder models. Note that we use StarCoder 15.5B and StarCoder 3B as teacher and student models, respectively.

| Methods | Python | | JAVA | | TypeScript | | C# | | Avg. | |
|---|---|---|---|---|---|---|---|---|---|---|
| | EM | ES | EM | ES | EM | ES | EM | ES | EM | ES |
| Teacher (15.5B) | 35.9 | 66.1 | 41.5 | 72.9 | 38.7 | 73.7 | 56.3 | 79.3 | 43.1 | 73.0 |
| Student (3B) | 20.8 | 41.5 | 25.3 | 51.4 | 25.7 | 56.2 | 40.5 | 60.5 | 28.1 | 52.4 |
| + CPT | 24.8 | 49.3 | 31.6 | 61.5 | 30.5 | 63.7 | 47.1 | 68.4 | 33.5 | 60.7 |
| + KD | 26.5 | 53.2 | 32.4 | 61.1 | 31.6 | 64.5 | 48.0 | 69.8 | 34.6 | 61.2 |
| + DDK (Ours) | 31.7 | 62.2 | 34.6 | 69.8 | 33.2 | 69.3 | 50.9 | 76.2 | 37.6 | 69.4 |

Table 6: Few-shot (**5-shot**) performance results of different methods on the Qwen-1.5 models. Note that we use Qwen-1.5 14B and Qwen-1.5 1.8B as teacher and student models, respectively.

| Methods | CEval | MMLU | GSM8K | Arc-E | Arc-C | Avg. |
|---|---|---|---|---|---|---|
| Qwen-14B | 79.86 | 66.30 | 69.14 | 89.24 | 82.25 | 77.36 |
| Student (1.8B) | 61.96 | 45.59 | 38.4 | 72.16 | 52.11 | 54.04 |
| + CPT | 60.92 | 45.60 | 43.36 | 73.10 | 52.28 | 55.05 |
| + KD | 61.66 | 44.28 | 50.26 | 73.87 | 54.69 | 56.95 |
| + DDK (Ours) | 65.38 | 47.59 | 55.19 | 76.64 | 57.01 | **60.36** |

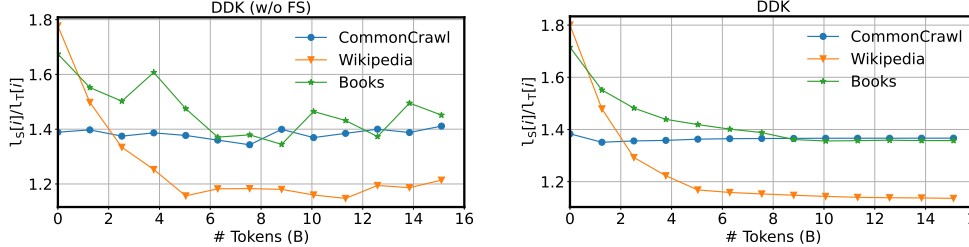

Figure 5: Visualization on the domain discrepancy among three domains.

**Visualization.** To better show the effectiveness of the factor smooth strategy in DDK, we compare the DDK (w/o FS) with our DDK by showing the domain discrepancy in the training process, where DDK (w/o FS) means that we remove the factor smooth updating strategy. Specifically, in Fig. 5, we compute the $(\ell_S[i]/\ell_T[i])$ as the ratio to represent the domain discrepancy for $i$-th domain, where a large ratio means a large discrepancy. As shown in Fig. 5, we observe that the ratio updates smoothly in DDK. Besides, in Table 4, the DDK is better than DDK (w/o FS), which means DDK can benefit a lot when using the factor smooth updating strategy.

Moreover, we refer readers to see Appendix B.2 and Appendix C for more details on the training costs and inference examples.

# 5    Conclusion

In this study, we introduce DDK, a novel framework for knowledge distillation tailored for LLMs. Our initial investigations underscore the criticality of optimizing domain data mixtures in the context of LLM distillation. To address this, we propose a domain knowledge-guided sampling approach that dynamically modulates the sampling probabilities across various domains. Furthermore, we put forward a factor smooth update strategy aimed at enhancing both the stability and the efficacy of the distillation process. Comprehensive evaluations of several benchmark datasets with diverse teacher-student model configurations demonstrate the effectiveness of the DDK framework.

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

# A  Broader Impacts and Limitations

**Broader Impacts.** Based on our DDK, we can boost the performance of small LLMs a lot under the guidance of the teacher LLMs. In real-world scenes, we hope our enhanced small LLMs can be deployed on low-resource devices (e.g., PCs or Mobiles) and make future life better with better small LLMs.

**Limitations.** First, we have to tune several hyperparameters to achieve better performance in our DDK. Second, due to the GPU resource limitations, we only distill the 1.1B, 1.8B and 4B sizes under the guidance of teachers with 7B and 10B+ sizes. In the future, we will investigate the performance of knowledge distillation using larger student and larger teacher models.

# B  More Details

## B.1  Discussion on the differences between DDK and Sheared LLaMA

First, for the solved tasks. the Sheared LLaMA [66] is proposed for structure pruning, while our DDK is proposed for knowledge distillation on the LLMs. Therefore, our DDK is orthogonal to the Sheared LLaMA and we can further improve the performance of small models pruned by Sheared LLaMA. Second, for the technical details, the sheared LLaMA needs to first fit a scaling function using three open-sourced LLaMA models (i.e., LLAMA2-7B/13B/70B) when pruning the LLaMA model series, where the fitting data points are very limited. In Sheared LLaMA, the authors also claim the estimated reference losses for different domains are also biased as the limited data points for estimating the scaling law parameters. Besides, in many real-world pruning scenarios, we cannot usually obtain a series of models under the same training setting. Thus, we cannot predict the losses across different domains of the small-size model when a series of models are not provided. In contrast, in DDK, we directly use the domain loss predicted by the teacher model as the reference loss without fitting the scaling function, where the teacher-predicted losses are used as the accurate guidance to improve the student model. Besides, the Sheared LLaMA is proposed to enhance the training efficiency of the continued pre-training setting of the pruned model, the guidance of the teacher model is not included. In addition, our DDK also introduces the factor smooth updating strategy, which is not used in Sheared LLaMA and can make the change of the domain mixture more stable. Moreover, in the following table, we also provide the results of replacing our proposed factor smooth updating strategy with the strategy of Sheared LLaMA, and observe that DDK is better a lot, which further shows the effectiveness of our factor smooth updating strategy.

## B.2  Details on the training costs

As shown in Table 7, we compare the TFLOPs of three representative baseline methods, and observe that the training costs of our DDK are acceptable when compared with the baseline KD method.

Table 7: Training TFLOPs on all data of different methods for Qwen-1.5. For KD and DDK, we use the Qwen-1.5 14B to distill the Qwen-1.5 1.8B.

| Models | **CPT** | **KD** | **DDK** |
|--------|---------|--------|---------|
| TFLOPs | 1.456e8 | 5.364e8 | 5.401e8 |

# C  More Sentence Examples

In the following, we provide more examples generated by the original and distilled models. We find that the sentences generated by the distilled model are superior to those generated by the original model in terms of fluency, relevance, and informativeness regarding the given topic.

## GSM8K

Prompt:
Question: Janet's ducks lay 16 eggs per day. She eats three for breakfast every morning and bakes muffins for her friends every day with four. She sells the remainder at the farmers' market daily for $2 per fresh duck egg. How much in dollars does she make every day at the farmers' market?
Answer: 18
Question: A robe takes 2 bolts of blue fiber and half that much white fiber. How many bolts in total does it take?
Answer: 3
Question: Josh decides to try flipping a house. He buys a house for $80,000 and then puts in $50,000 in repairs. This increased the value of the house by 150%. How much profit did he make?
Answer: 70000
Question: James decides to run 3 sprints 3 times a week. He runs 60 meters each sprint. How many total meters does he run a week?
Answer:
Assistant (Qwen-1.5 1.8B):
1800
Assistant (Qwen-1.5 1.8B DDK):
540
Prompt:
Gunter is trying to count the jelly beans in a jar. He asks his friends how many they think are in the jar. One says 80. Another says 20 more than half the first one. A third says 25% more than the first one. What is their average guess?
Answer:
Assistant (Qwen-1.5 1.8B):
First, we need to find half of the first number, which is 80 / 2 = 40.
Then, we need to add 20 more to that number, which is 40 + 20 = 60.
Next, we need to find 25% of the first number, which is 80 * 0.25 = 20.
Finally, we add that number to the first number, which is 60 + 20 = 80.
Now, we can find the average by adding up all the guesses and dividing by the number of guesses, which is 80 + 60 + 20 = 160.
Finally, we divide that number by the number of guesses, which is 3, to find the average, which is 160 / 3 = 53.33.
Therefore, the answer is 53.
Assistant (Qwen-1.5 1.8B DDK):
The first friend's guess is 80 jelly beans.
The second friend's guess is 20 more than half of 80, which is (80/2) + 20 = 40 + 20 = 60 jelly beans.
The third friend's guess is 25% more than 80, which is 80 + (0.25 * 80) = 80 + 20 = 100 jelly beans.
To find the average guess, we add up all the guesses and divide by the number of friends, which is 3.
So the average guess is (80 + 60 + 100) / 3 = 240 / 3 = 80 jelly beans.
#### 80
The answer is: 80

Prompt:
问题：冬瓜、黄瓜、西瓜、南瓜都能吃，什么瓜不能吃？
答案：
Assistant (Qwen-1.5 1.8B):
冬瓜、黄瓜、西瓜、南瓜都能吃，南瓜不能吃。
Assistant (Qwen-1.5 1.8B DDK):
傻瓜
解析：冬瓜、黄瓜、西瓜、南瓜都能吃，傻瓜不能吃。
Prompt:
问题：什么东西天气越热，它爬的越高？
答案：温度计
问题：什么动物，你打死了它却流了你的血？
答案：蚊子
问题：什么布剪不断？
答案：
Assistant (Qwen-1.5 1.8B):
蚊帐
Assistant (Qwen-1.5 1.8B DDK):
瀑布
Prompt:
如何写一篇关于环保的激励人心的演讲稿？：
Assistant (Qwen-1.5 1.8B):
环保是当今社会面临的一个重要问题，它不仅关系到我们的生存环境，也关系到我们的未来。因此，我们需要采取行动来保护我们的地球。以下是一篇关于环保的激励人心的演讲稿：
尊敬的各位领导、各位来宾、亲爱的朋友们：
大家好！今天，我非常荣幸能够在这里发表关于环保的演讲。环保是我们每个人都应该关注的问题，它不仅关系到我们的生存环境，也关系到我们的未来。
首先，让我们来看看环保的重要性。我们的地球是我们的家园，它是我们生存的基础。然而，由于人类的过度开发和污染，我们的地球正面临着严重的环境问题。例如，全球变暖、气候变化、水资源短缺、土地退化等问题，这些问题不仅影响着我们的健康和生活质量，也威胁着我们的生存环境。
因此，我们必须采取行动来保护我们的地球。我们可以从以下几个方面入手：
首先，我们应该减少对环境的污染。我们应该尽可能地减少使用一次性塑料制品，如塑料袋、塑料瓶等。我们应该尽量使用可重复使用的物品，如购物袋、水瓶等。此外，我们还可以选择使用环保产品，如环保清洁剂、环保洗衣液等。
其次，我们应该节约能源。我们应该尽可能地减少能源的消耗，如
Assistant (Qwen-1.5 1.8B DDK):
好的，以下是一篇关于环保的激励人心的演讲稿：尊敬的各位来宾，大家好！
今天，我想和大家分享一些关于环保的话题。我们都知道，环保是我们每个人都应该关注的问题。我们的地球是我们共同的家园，我们应该尽我们所能来保护它。
首先，我们需要认识到环保的重要性。我们的地球正面临着许多环境问题，如气候变化、空气污染、水污染等等。这些问题不仅影响着我们的健康和生活质量，也影响着我们的未来。因此，我们必须采取行动来保护我们的地球。
其次，我们需要采取行动来保护我们的环境。我们可以从小事做起，如节约用水、节约用电、减少使用塑料袋等等。我们还可以参加环保活动，如植树造林、垃圾分类等等。这些行动虽然看似微不足道，但它们可以积少成多，对环境产生积极的影响。
最后，我们需要鼓励更多的人加入到环保行动中来。我们可以向身边的人宣传环保知识，鼓励他们采取环保行动。我们还可以加入环保组织，参与环保活动，为环保事业贡献自己的力量。
尊敬的各位来宾，让我们一起行动起来，保护我们的地球，让我们的家园更加美好！谢谢大家！

