# OpenReview forum: "DDK: Distilling Domain Knowledge for Efficient Large Language Models"
_NeurIPS.cc/2024/Conference — NeurIPS 2024 poster_

### Official Review · Reviewer_6kDi · 2024-07-07

**Soundness:** 2
**Presentation:** 3
**Contribution:** 2
**Rating:** 6
**Confidence:** 4

**Summary:**

This paper proposes DDK, a knowledge distillation (KD) framework that distills large language models (LMs) into small LMs. Unlike previous KD methods, DDK dynamically adjusts the domain weights during distillation. Experiments show that DDK outperforms other KD baselines across various tasks.

**Strengths:**

1. The paper is well written and the method is easy to follow.
2. The experiments show that DDK outperforms other KD baselines on various tasks.

**Weaknesses:**

The extra computation introduced by KKD should be considered. It seems KKD requires the inference of a large LM during the training of the small LM. When the teacher model is much larger than the student model (QWen-1.5 14B v.s. QWen-1.5 1.8B), the inference cost of the teacher model would be even larger than training the student model. Therefore, it is more reasonable to compare the performance of the distilled model and the baselines given the same FLOPs.

**Questions:**

1. What are the training data for the baselines (CPT, TED, KD, and MiniLLM)? Is the data for DDK the same as that for the baseline methods?
2. In lines 178-179, is the learning rate 3e-5 ($3\times 10^{-5}$) rather than $3e^{-5}$?

---

> ### Author Rebuttal · Authors · 2024-08-06
>
> Thanks for your careful reading and constructive suggestions. We will address your concerns shown below in detail.
>
> **Q1: Extra computation introduced by DDK should be considered. Compare the performance of the distilled model and the baselines given the same FLOPs.**
>
> **A1**: Thanks for your insightful suggestions. In Table 13 of Appendix B.2, we also report the training costs (i.e., TFLOPs) of different baseline methods, and we observe that the additional computation costs are acceptable when compared with the baseline KD method. Besides, we observe the  ratio between the TFLOPs of DDK and  TFLOPs of CPT is (5.401e8/1.456e8)=3.709. Then, we also perform the experiments by continuing  pre-training the student  model with about 56B tokens, where the TFLOPs is about 5.4e8. The results are shown in the following table. In the table, we observe that when we continue pretraining the student model for more tokens without using the teacher guidance, little gain is obtained. We assume that the Qwen1.5 model has been trained on >3T tokens, and the convergence status is stable when using the standard pretraining method. In contrast, when using the teacher guidance of the DDK, we observe consistent  performance improvements in Table 1, Table 2, Table 7, and Table 8, which further demonstrates the effectiveness of our DDK.
>
> | Models                | TFLOPs  | CEval | MMLU  | RACE  | C3    | W.G.  | GSM8K | C.QA  | Arc-E | Arc-C | H.E.  | MBPP  | Avg.  |
> | --------------------- | ------- | ----- | ----- | ----- | ----- | ----- | ----- | ----- | ----- | ----- | ----- | ----- | ----- |
> | Teacher (Qwen1.5-14B) | -       | 78.68 | 64.34 | 89.95 | 77.38 | 68.74 | 67.63 | 82.06 | 87.58 | 80.59 | 37.80 | 44.00 | 70.80 |
> | Student (Qwen1.5-1.8B)        | -       | 59.66 | 44.48 | 69.57 | 58.27 | 57.85 | 38.4  | 64.70 | 70.23 | 50.31 | 11.87 | 18.00 | 49.39 |
> | +CPT (15B tokens)     | 1.456e8 | 60.13 | 45.01 | 69.00 | 60.30 | 56.98 | 42.50 | 64.78 | 72.00 | 51.03 | 13.12 | 20.45 | 50.48 |
> | +DDK (15B toekns)            | 5.401e8 | 63.75 | 46.01 | 71.56 | 65.53 | 59.10 | 53.54 | 66.75 | 75.01 | 55.03 | 27.13 | 26.10 | 55.41 |
> | +CPT (56B tokens)     | 5.436e8 | 60.15 | 45.00 | 69.16 | 60.43 | 57.06 | 42.91 | 64.86 | 72.14 | 51.19 | 13.41 | 20.74 | 50.64 |
>
> We will revise the above discussion in our new version.
>
> **Q2: Is the data for DDK the same as that for the baseline methods?**
>
> **A2**: The training data is the same and we adopt the same seed for shuffling the dataset, which can ensure the training data  same for different methods. We will clarify this detail in our new version.
>
> **Q3: Learning rate issue.**
>
> **A3**: Thanks for your kind suggestions. We will revise this typo in the new version.

---

> > ### Comment · Reviewer_6kDi · 2024-08-09
> > **Official Comment by Reviewer 6kDi**
> >
> > I thank the authors for their response. After reading the response, I think my current score is appropriate.

---

> > > ### Author Response · Authors · 2024-08-10
> > >
> > > Thanks for your feedback. We will carefully address your concerns in our new version.

---

### Official Review · Reviewer_oZHE · 2024-07-11

**Soundness:** 3
**Presentation:** 3
**Contribution:** 2
**Rating:** 6
**Confidence:** 5

**Summary:**

The paper introduces a new framework called Dynamic Domain Knowledge Distillation (DDK) to enhance the efficiency of knowledge distillation for large language models (LLMs). Unlike traditional methods that overlook domain performance differences between student and teacher models, DDK dynamically adjusts the distillation dataset composition based on these differences, ensuring a more stable and effective knowledge transfer. This approach addresses the issue of excessive focus on domains with minimal performance gaps and enhances overall model performance. Extensive evaluations demonstrate that DDK significantly outperforms existing knowledge distillation methods and continuously pretrained baselines.

**Strengths:**

- The proposed dynamic dataloader for KD is technically sound.
- Numerical experiments well validate the efficacy of the method.

**Weaknesses:**

- Dynamic dataloader requires knowing the training data distribution and category beforehand.
- Missing references. Similar ideas have been explored in pruning LLMs, such as ShearedLLaMA, LoRAShear to recover the knowledge . The paper needs to discuss with them in the related work section due to the closed relation between pruning and KD.

Sheared LLaMA: Accelerating Language Model Pre-training via Structured Pruning

LoRAShear: Efficient Large Language Model Structured Pruning and Knowledge Recovery

**Questions:**

- How the methods perform on under other KD losses, such as reversed KLD, JSD, skew-KLD.

On-policy distillation of language models: Learning from self-generated mistakes.

DistiLLM: Towards Streamlined Distillation for Large Language Models.

**Limitations:**

Yes.

---

> ### Author Rebuttal · Authors · 2024-08-06
>
> Thanks for your careful reading and constructive suggestions. We will address your concerns shown below in detail.
>
> **Q1: Require knowing the training data distribution and category beforehand.**
>
> **A1**: Our DDK requires the training data distribution and category to divide the domains and we acknowledge this limitation. However, it should be mentioned that existing pretraining datasets usually provide the source composition. For example, the Repajama[https://www.together.ai/blog/redpajama] includes 7 domains (i.e.,  CommonCrawl, C4, Github, Wikipedia, Books, ArXiv, and StackExchange), The Pile [https://arxiv.org/pdf/2101.00027] includes 22 domains (i.e., Pile-CC, PubMed Central, Books3, OpenWebText2 and etc.) and Dolma[https://arxiv.org/pdf/2402.00159] includes 6 domains (i.e., Common Crawl, GitHub, Reddit, Semantic Scholar and etc.)
>
> Meanwhile, we also tried another option to dive the training data into 10 domains based on the clustering method to address this limitation, where we first extract the feature of each document using [https://huggingface.co/BAAI/bge-m3] and then use  k-means  to divide the training corpus into 10 domains implicitly. Then, we apply DDK on  the divided implicit domains based on the clustering and we call this method  DDK (clustering), where we do not need to know the training data distribution and category. In the following table, we observe that the DDK (clustering) achieves relatively comparable performance with original DDK. We assume that the explicitly divided domains based on the sources (e.g., The stack, OpenWebMath) are distinct. Besides, we will continue to investigate the clustering method, which is more scalable.
>
> | Models                        | CEval | MMLU  | RACE  | C3    | W.G.  | GSM8K | C.QA  | Arc-E | Arc-C | H.E.  | MBPP  | Avg.  |
> | ----------------------------- | ----- | ----- | ----- | ----- | ----- | ----- | ----- | ----- | ----- | ----- | ----- | ----- |
> | Student (1.8B)+DDK            | 63.75 | 46.01 | 71.56 | 65.53 | 59.1  | 53.54 | 66.75 | 75.01 | 55.03 | 27.13 | 26.1  | 55.41 |
> | Student(1.8B)+DDK(clustering) | 63.19 | 46.11 | 72.41 | 67.03 | 59.03 | 51.59 | 65.01 | 74.91 | 55.95 | 25.43 | 26.38 | 55.19 |
>
> **Q2: Cite and discuss shearllama and lora shear.**
>
> **A2:**  Please See **General Response** (**G.Q2**) on the discussion between Sheared LLaMA and DDK.
>
> The discussion between LoRAShear and DDK is as follows.
>
> The LoraShear is proposed for structured pruning on LLMs, which first creates the dependency graphs over LoRA module and then proceeds progressive structured pruning on LoRA adaptors.  To recover the lost knowledge during pruning, LoRAShear proposes a Dynamic Knowledge Recovery scheme on both the pre-training and instructed fine-tuning datasets to effectively narrow down the performance gap to the full models.
> In contrast to  LoraShear, DDK aims to efficiently transfer the domain knowledge of the teacher network, and propose a factor smooth updating strategy to stabilize the domain knowledge guided sampling process.
>
> We will cite these works in our new version.
>
> **Q3: How the methods perform under other KD losses (reversed KLD, JSD, skew-KLD)?**
>
> **A3:** To address the mode-averaging problem of the student model, where the student model learns an overly smooth distribution in an attempt to cover the teacher’s entire support set, the reversed KLD in MiniLLM and generalized GSD in  [R1]  are proposed. However, these on-policy approaches (i.e.,  the reversed KLD and the generalized JSD) show lower efficiency, the skew-KLD in [R2] introduces the adaptive off-policy approach to enhance the efficiency. In contrast to these works, our DDK focuses on efficiently transferring the domain knowledge of the teacher network based on standard forward KD loss, where the motivation and technical details are different from these works a lot. Besides, our DDK is also orthogonal to these KD losses, where we can directly replace the KL divergence in Eq. 3 with the above losses in MiniLLM, [R1] and [R2] easily. In the following table, we report the results of our DDL using different distillation losses, and we observe that these losses have not provided more performance gains for our DDK based on KLD. For this phenomenon, we assume that our DDK focuses on the pre-training setting with a relatively large training data set, where the data quality and data mixture are very important for pretraining (See **G.Q1**). Meanwhile, in MiniLLM and [R1, R2], these losses mainly focus on the SFT phase to improve learning efficiency by addressing the mode-averaging problem. Therefore, we assume that the main contribution on the improvements comes from changing the domain mixture in our DDK. In the future, we will also continue to investigate the optimal distillation loss formulation for our DDK following [R1, R2].
>
> | Models                         | CEval | MMLU  | RACE  | C3    | W.G.  | GSM8K | C.QA  | Arc-E | Arc-C | H.E.  | MBPP  | Avg   |
> | ------------------------------ | ----- | ----- | ----- | ----- | ----- | ----- | ----- | ----- | ----- | ----- | ----- | ----- |
> | Student (Qwen1.5-1.8B)         | 59.66 | 44.48 | 69.57 | 58.27 | 57.85 | 38.4  | 64.7  | 70.23 | 50.31 | 11.87 | 18    | 49.39 |
> | +DDK (Default KLD in Eq.3)     | 63.75 | 46.01 | 71.56 | 65.53 | 59.1  | 53.54 | 66.75 | 75.01 | 55.03 | 27.13 | 26.1  | 55.41 |
> | +DDK (Reversed KLD in MiniLLM) | 62.19 | 45.85 | 70.66 | 65.03 | 54.67 | 50.6  | 65.11 | 74.51 | 52.38 | 23.44 | 24.67 | 53.56 |
> | +DDK(JSD in [R1])              | 62.63 | 45.72 | 71.7  | 65.58 | 57.14 | 53.07 | 67.08 | 75.34 | 54.95 | 22.61 | 25.15 | 54.63 |
> | +DDK(skew-KLD in [R2])         | 62.96 | 45.42 | 71.84 | 65.71 | 58.59 | 53.45 | 67.16 | 75.46 | 55.03 | 26.05 | 25.98 | 55.24 |
>
> [R1]: On-policy distillation of language models: Learning from self-generated mistakes, https://arxiv.org/pdf/2306.13649
>
> [R2]: DistiLLM: Towards Streamlined Distillation for Large Language Models, https://arxiv.org/pdf/2402.03898

---

> > ### Comment · Reviewer_oZHE · 2024-08-10
> >
> > I am satisfied with the additional discussions and results, upon which I decide to increase my rate.

---

> > > ### Author Response · Authors · 2024-08-11
> > >
> > > Thanks for your feedback. We will carefully address your concerns in our new version.

---

### Official Review · Reviewer_EYh3 · 2024-07-18

**Soundness:** 2
**Presentation:** 2
**Contribution:** 2
**Rating:** 5
**Confidence:** 4

**Summary:**

The work introduces a novel framework for knowledge distillation (KD) for LLMs. The key innovation of DDK is its dynamic adjustment of the distillation dataset composition based on domain performance differences between the teacher and student models. The paper presents extensive evaluations demonstrating that DDK significantly improves performance in various KD settings, outperforming both continuous training baselines and existing KD methods.

**Strengths:**

1.  The authors provide extensive empirical evidence demonstrating the effectiveness of DDK in improving the performance of student models across various benchmarks.
2.  As the computational and storage demands of LLMs are significant barriers to their widespread deployment, KD is a promising solution.  The proposed KD method is simple and easy to follow.

**Weaknesses:**

1. Discuss the difference between DDK and the Dynamic Batch Loading proposed by Sheared LLaMA[1], which is also
 proposed to adjust domain proportions for dynamically training smaller models. They also identify discrepancies in loss between smaller and larger models across various domains, and accordingly, they sample more data from domains where the discrepancy is more pronounced. While they concentrate on structural pruning, it is akin to the DDK. Consequently, I perceive the novelty of DDK as being somewhat limited.
2. The results of Qwen 1.5 in Table 1 are not significantly convincing. The MMLU/HumanEval of Qwen 1.5 1.8B in the Qwen official blog are  46.8/20.1 while the authors' report is 44.5/11.9. In addition, compared to the official results, we can see that the DDK fails to improve the model of the students on MMLU. The authors need to check this and provide **more robust results of baselines**.

[1] SHEARED LLAMA: ACCELERATING LANGUAGE MODEL PRE-TRAINING VIA STRUCTURED PRUNING. Xia et al., 2023

**Questions:**

1. In the setting of paper, the domains are predefined. How to extend the DDK framework for new domain during the distillation training process? Could give more experiments on continual domain learning settings?

**Limitations:**

Yes.

---

> ### Author Rebuttal · Authors · 2024-08-06
>
> Thank you for your valuable comments.
>
> **Q1: Discuss with Sheared LLaMA. Novelty.**
>
> **A1:** Please See **General Response** (**G.Q1** and **G.Q2**).
>
> **Q2: Qwen1.5 results on MMLU and Humaneval.**
>
> **A2:** For MMLU, the accuracy from Qwen Blog is based on a 5-shot setting, while the result in Table 1 is based on a zero-shot setting (See Line 182). Besides, we provide the 5-shot result of MMLU in Table 12 and observe the result is 45.59, which is close to the provided result (46.8) of the Qwen blog. Besides, we assume the gap between 46.8 and 45.59 comes from the evaluation prompt. Specifically,  the evaluation prompt for DDK is as follows:
>
> ```SQL
> # Prompt-1
> for opt in ['A', 'B', 'C', 'D']:
>     prompt="Answer the question.\nQuestion: {question}\nChoices:\nA. {A}\nB. {B}\nC. {C}\nD. {D}\nAnswer: " + opt
> ```
>
>  The ``{question}``, ``{A}``, ``{B}``, ``{C}``, ``{D}`` are the corresponding inputs for each evaluation sample.
>
> We also have tried the following prompt from OpenCompass [https://github.com/open-compass/opencompass], where the task description is also added to the evaluation prompt. Note that the ``mmlu_all_sets`` contains 57 tasks. Based on this prompt, we obtain 46.88 and 47.05 under the zero-shot and few-shot settings, respectively.
>
> ```SQL
> # Prompt-2
> mmlu_all_sets = ['college biology', 'college chemistry', 'college computer science', ..., 'human aging', 'us foreign policy', 'conceptual physics']
> for task in mmlu_all_sets:
>     for opt in ['A', 'B', 'C', 'D']:
>         prompt="The following are multiple choice questions (with answers) about" + task + '\n\n' + "{question}\nA. {A}\nB. {B}\nC. {C}\nD. {D}" + '\n' + 'Answer: ' + opt
> ```
>
>  We also provide the following results and DDK achieves consistent improvements in MMLU.
>
> |                | Prompt-1 | Prompt-2 | Prompt-1(few-shot) | Prompt-2(few-shot) |
> | -------------- | -------- | -------- | ------------------ | ------------------ |
> | Student (1.8B) | 44.48    | 46.88    | 45.59              | 47.05              |
> | +DDK           | 46.01    | 47.95    | 47.59              | 48.12              |
>
> For Humeneval, we observe that the results of Qwen1.5-1.8B  are very sensitive to the prompt, and we use the naive testing prompt (accuracy  of 11.87) as follows:
>
> ```SQL
> # Prompt-1
> prompt='{prompt}'
> ```
>
> Similarly, we have tried following two prompts following OpenCompass:
>
> ```SQL
> # Prompt-2
> prompt='Complete the following python code:\n{prompt}'
> # Prompt-3
> prompt='You are an intelligent programming assistant to produce Python algorithmic solutions.\nCan you complete the following Python function?\n```python\n{prompt}\n```'
> ```
>
> And we observe the results of ``Prompt-2`` and ``Prompt-3`` are 23.17 and 26.83, respectively, which are higher than the reported result (20.1) on Qwen blog.
>
> Meanwhile, we report the DDK results on these three evaluation prompt settings. In the following table, we observe that DDK still achieves better performance than the baseline. In new version, we will provide a detailed analysis of the evaluation prompt and provide more solid and robust results.
>
> |                | Prompt-1 | Prompt-2 | Prompt-3 |
> | -------------- | -------- | -------- | -------- |
> | Student (1.8B) | 11.87    | 23.17    | 26.83    |
> | +DDK           | 27.13    | 29.59    | 31.08    |
>
> **Q3: Extend DDK for new domains during distillation.**
>
> **A3:** In distillation, when a new domain is introduced, we first need to prepare the validation dataset for this domain following Appendix B.1 (Line 490-510). Then, following Eq. (1),  based on the current model checkpoint, we compute the perplexity scores over the validation sets of all domains for student and teacher respectively, and obtain the newly initialized domain discrepancy factor, which is then used to change the domain mixture following the Algorithm 1  of the main paper.
>
> **Q4: Experiments on continual domain learning setting.**
>
> **Q4:** We provide the results of involving a new source domain data (i.e., Law) using the Pile-of-Law dataset [https://huggingface.co/datasets/pile-of-law/pile-of-law]. Specifically, we first train the original 10 domains for 5B/10B tokens. Then, we append the Law domain into the training dataset. After that,  following Algorithm 1 of the main paper, we continue the distillation training on the student for additional 10B/5B tokens on these 11 domains, which aims to achieve a fair comparison with the DDK baseline results on 10 domains for about 15B tokens.  To show the effect of introducing the law domain, we additionally report the results on the law subset of MMLU-Pro[https://huggingface.co/datasets/TIGER-Lab/MMLU-Pro]. (MMLU-Pro has 14 high-quality subsets with challenging questions.), and observe that by introducing the law domain, better results on the MMLU-Pro(Law) are obtained, and results on other datasets are also preserved well.
>
> | Models                                                       | MMLU  | RACE  | W.G.  | GSM8K | Arc-E | Arc-C | MMLU-Pro(Law) | Avg.  |
> | ------------------------------------------------------------ | ----- | ----- | ----- | ----- | ----- | ----- | ------------- | ----- |
> | Student (Qwen1.5-1.8B)                                       | 44.48 | 69.57 | 57.85 | 38.4  | 70.23 | 50.31 | 14.4          | 49.32 |
> | Student+DDK(train 5B on original 10 domains)                 | 44.12 | 69.79 | 58.17 | 44.35 | 72.39 | 52.75 | 14.8          | 50.91 |
> | Student+DDK(train 5B on original 10 domains+10B with additional new law domain) | 45.98 | 71.43 | 58.61 | 51.12 | 74.78 | 54.95 | 16.29         | 53.31 |
> | Student+DDK(train 10B on original 10 domains)                | 45.85 | 70.96 | 59.19 | 49.66 | 74.07 | 53.18 | 14.68         | 52.51 |
> | Student+DDK(train 10B on original 10 domains+5B with additional new law domain) | 46.12 | 71.02    | 58.8  | 52.51 | 74.36 | 54.27 | 15.75         | 53.26 |
> | Student+DDK(train 15B on original 10 domains, baseline in Table 1) | 46.01 | 71.56 | 59.1  | 53.54 | 75.01 | 55.03 | 14.74         | 53.57 |

---

> > ### Author Response · Authors · 2024-08-12
> > **Looking forward to Feedback as Discussion Deadline Approaches**
> >
> > Hi, we sincerely thank you very much for these constructive comments and evaluation of our manuscript. As the discussion phase will be closed on Aug 13 11:59 pm AoE, we would like to kindly ask you to take a look at our responses and reevaluate our work based on our clarifications. Please let us know whether our response addresses your concerns or whether there is any further detail we can provide to help address these concerns.
> >
> > Thank you again for dedicating your time to reviewing our paper.

---

> > > ### Comment · Reviewer_EYh3 · 2024-08-13
> > >
> > > My main concerns and questions are set out in the rebuttal. I suggest that the author add a comparison or discussion of **DYNAMIC BATCH LOADING** in sheared llama in the next version, as I still think they are similar techniques.

---

> > > > ### Author Response · Authors · 2024-08-13
> > > >
> > > > Thanks for your feedback. We will carefully address your concerns and questions in our new version.

---

### Official Review · Reviewer_foae · 2024-07-26

**Soundness:** 3
**Presentation:** 3
**Contribution:** 2
**Rating:** 5
**Confidence:** 4

**Summary:**

This work proposed a KD strategy for LLMs. Specifically, with assess to the domain-specific performance of both the teacher and student LLMs, DDK uses domain knowledge guided sampling to dynamically update the data mixture. In addition the paper also conducts a statistical analysis of the domain distribution of the datasets involved. The training process is relatively straightforward and easy to generalize. The experimental results also show that DDK's training method improves the performance average of different data sets.

**Strengths:**

1. A complete training algorithm is designed, and the process is explained clearly. The process of the DDK algorithm is easy to extend to the training process of other models.
2. The authors conducted a comprehensive knowledge distillation experiment on two large model families and a comprehensive ablation study.

**Weaknesses:**

1. Although the method proposed in this paper is easy to understand and effective, I doubt that the method in this paper is limited to LLMs. In other words, this paper does not mention (or needs to explain) how previous researchers (before LLMs) performed domain-enhanced distillation for domain-biased datasets, and why these previous methods cannot be applied to the distillation of LLMs to achieve similar results. The advantages and novelty of this paper's domain sampling method over previous work that may be transferable to LLMs need further explanation.
2. In the experimental part, there is a lack of key comparison between DDK and other methods that focus on similar domain sampling. The baseline actually involves the work that focuses on domain in KD (cited as [60], etc.), but the subsequent analysis only compares the total average score of DDK and these works, which seems to lack comparison and analysis of similar works. As far as I know, other baselines are more general KDs, and do not focus on domain information.
It is certainly worth noting that DDK performs better than baselines such as MiniLLM, but I think what can better illustrate the effectiveness and novelty of this paper is the comparison with similar domain data sampling, including experimental analysis.
3. In the experimental section, you can add experiments on the dataset and the scale property of the teacher model. This is a possible suggestion.

**Questions:**

The questions I expect to ask would be similar to the above section.

**Limitations:**

The method proposed in this paper is very complete and solves the problem well, but there are perhaps two points to note:
1. The distillation method designed in this paper does not seem to be necessarily limited to LLMs. One of the main difficulties of KD on LLM may be that the distribution difference between the teacher and student models is too large, but the key points that this paper focuses on and solves seem to be orthogonal to this point. So it also leads to a similar question: why previous similar methods that focus on domain sampling cannot be migrated to LLMs, and what are the advantages and novelties of this paper's design.
2. Following point 1, in the experiment, what are the specific advantages of this method over the predecessors in the domain problem (not just the overall average, of which result shows DDK outperformed others). This may be what I am very curious about after reading the paper.

---

> ### Author Rebuttal · Authors · 2024-08-06
>
> Thank you for your nice comments and suggestions.
>
> **Q1: Compare domain-enhanced KD methods.**
>
>  **A1:** After investigating KD [R1] and applications on LLMs [R2], we have observed existing domain-enhanced KD methods can be divided into two categories. The first is cross-domain KD for domain adaptation to improve the generalization abilities of the unseen target domains based on seen source domains. In contrast, DDK aims to improve training efficiency and effectiveness given multiple seen domains, and these cross-domain KD methods cannot be applied to LLM pretraining. The second is based on multi-teacher strategy [R3, R4], which first trains different domain expert models and then distills the overall domain capacities into one single model. Thus, we implement two baselines following [R3, R4].
>
>  [R1]. Knowledge Distillation: A Survey
>
>  [R2]. A Survey on Knowledge Distillation of Large Language Models
>
>  [R3]. BAM! Born-Again Multi-Task Networks for Natural Language Understanding
>
>  [R4]. Knowledge Fusion of Large Language Models
>
>  Specifically, first, we train the domain expert models on 10 different domains as teacher models, where each domain expert is trained on about 5B tokens on the corresponding domain. Then, for the first baseline [R3], we select a corresponding teacher to produce logits for each sample based on the belonged domain. For the second baseline [R4], we ensemble logits from different domain teacher models. The table below lists the PPL on validation datasets for 10 domains.
>
> |         | CC    | C4    | The Stack | Wikipedia | Books | Arxiv | StackExchange | Chinese Books | Chinese CC | OpenWebMath |
> | ------- | ----- | ----- | --------- | --------- | ----- | ----- | ------------- | ------------- | ---------- | ----------- |
> | Student | 10.15 | 18.50 | 4.14      | 13.27     | 25.81 | 12.97 | 10.04         | 23.54         | 21.20      | 9.50        |
> | +[R3]   | 10.00 | 19.14 | 4.14      | 10.12     | 22.73 | 12.90 | 8.37          | 20.59         | 20.19      | 4.75        |
> | +[R4]   | 10.06 | 19.09 | 4.38      | 9.54      | 26.01 | 12.72 | 9.46          | 20.51         | 20.12      | 5.97        |
> | +DDK    | 10.02 | 18.94 | 3.95      | 8.36      | 20.43 | 11.23 | 7.22          | 19.69         | 20.33      | 4.62        |
>
>  We also report the downstream results and observe that baseline methods are inferior to DDK. We assume that these methods only consider producing better teacher guidance, and do not consider the effect of the domain mixture for  LLM pretraining.
>
> | Models  | MMLU  | RACE  | W.G.  | GSM8K | Arc-E | Arc-C | H.E.  | Avg.  |
> | ------- | ----- | ----- | ----- | ----- | ----- | ----- | ----- | ----- |
> | Student | 44.48 | 69.57 | 57.85 | 38.4  | 70.23 | 50.31 | 11.87 | 48.96 |
> | +[R3]   | 45.02 | 68.25 | 58.8  | 47.24 | 71.17 | 52.44 | 19.09 | 51.72 |
> | +[R4]   | 45.53 | 68.32 | 58.25 | 39.65 | 74.33 | 53.84 | 20.26 | 51.45 |
> | +DDK    | 46.01 | 71.56 | 59.1  | 53.54 | 75.01 | 55.03 | 27.13 | 55.34 |
>
> **Q2: Experiments on dataset and scale property of teacher.**
>
> **A2:** For experiments on dataset, we have performed experiments using StarCoder on Stack V2 dataset in Table 11. Besides, we perform experiments on The Pile dataset [https://arxiv.org/pdf/2101.00027] with 22 domains, and we use Qwen-1.5 14B and Qwen-1.5 1.8B as teacher and student following Table 1. Note that as The Pile focuses on English, we do not evaluate CEval and C3.
>
> | Models            | MMLU  | RACE  | W.G.  | GSM8K | C.QA  | Arc-E | Arc-C | H.E.  | MBPP  | Avg.  |
> | ----------------- | ----- | ----- | ----- | ----- | ----- | ----- | ----- | ----- | ----- | ----- |
> | +CPT (main paper) | 45.01 | 69.00 | 56.98 | 42.50 | 64.78 | 72.00 | 51.03 | 13.12 | 20.45 | 48.32 |
> | +DDK (main paper) | 46.01 | 71.56 | 59.10 | 53.54 | 66.75 | 75.01 | 55.03 | 27.13 | 26.10 | 53.36 |
> | +CPT(The Pile)    | 44.74 | 68.86 | 56.43 | 40.56 | 64.29 | 71.13 | 50.34 | 12.20 | 18.48 | 47.45 |
> | +DDK(The Pile)    | 45.72 | 69.98 | 58.80 | 45.41 | 65.44 | 74.03 | 54.52 | 22.56 | 22.59 | 51.01 |
>
> We observe that DDK  achieves better performance than CPT baseline on The Pile and results on The Pile are relatively lower than results in our main paper. We assume that the data quality used in our main paper (See Line 166-171) is better than The Pile, as we directly use the original Pile dataset in 2021 without additional cleaning strategies.
>
> For scale property of teacher, we have used different sizes of teacher models (i.e., Qwen1.5-14B and Qwen1.5-7B) to distill Qwen1.5-1.8B in Table 1 and Table 8, and average results distilled by Qwen1.5-14B and Qwen1.5-7B are 55.41 and 53.64, respectively, which means that better results are achieved using a relatively larger teacher.
>
> **Q3: Advantages and novelties of DDK.**
>
> **A3:** Please See **General Response** (**G.Q1**).
>
> **Q4: Advantages over predecessors  (not the overall average).**
>
> **A4:** First, in Fig. 1, DDK can reduce the performance gaps between teacher and student a lot in many domains. For example, for DDK, PPL on the ``Books`` domain is less than other methods a lot, and PPL  on  ``StackExchange`` domain is very close to teacher.
>
> Second, in Table 1, we observe that a large relative performance gap usually exists in reasoning-related tasks for teacher and student. For example, relative performance gap ratios are 43.22% [(67.63-38.4)/67.63=43.22%] and 68.59%[(37.80-11.87)/37.80=68.59%] for GSM8K and Humeneval, respectively. In contrast, for knowledge-related tasks, the relative performance gap ratios are 24.17% [(78.68-59.66)/78.68=24.17%] and 30.87%[(64.34-44.48)/64.34=30.87%] for CEval and MMLU. After using DDK, the relative performance gaps are 20.84%, 28.23%, 18.98%, 28.49% on GSM8K, Humeneval, CEval and MMLU. In contrast, for ``CPT & DoReMi'' and MiniLLM, the relative performance gaps are [32.18%, 76.85%, 21.91%, 30.15%] and[27.69%, 55.34%, 21.63%, 29.95%], respectively. Thus, we assume that  DDK can greatly improve the weaknesses on reasoning-related tasks.

---

> > ### Author Response · Authors · 2024-08-12
> > **Looking forward to Feedback as Discussion Deadline Approaches**
> >
> > Hi, we sincerely thank you very much for these constructive comments and evaluation of our manuscript. As the discussion phase will be closed on Aug 13 11:59 pm AoE,  we would like to kindly ask you to take a look at our responses and reevaluate our work based on our clarifications. Please let us know whether our response addresses your concerns or whether there is any further detail we can provide to help address these concerns.
> >
> > Thank you again for dedicating your time to reviewing our paper.

---

> > ### Author Response · Authors · 2024-08-13
> > **Combine MiniLLM with the domain sampling method of DDK.**
> >
> > The comparison with MiniLLM in a similar domain sampling strategy is also discussed in **Q3** from **Reviewer oZHE**, where we can directly replace the KL divergence loss in Eq. 3 with the reversed KLD loss in MiniLLM. In **A3** for **Q3** from **Reviewer oZHE**, we call this alternative method as **DDK (Reversed KLD in MiniLLM)**, and we observe that replacing the distillation loss does not bring more gains when compared with the DDK with default KL divergence loss. For this phenomenon, just discussed in **A3** , we assume that our DDK focuses on the pre-training setting with a relatively large training data set, where the data quality and data mixture are very important for pretraining (See **G.Q1**). Meanwhile, the MiniLLM focuses on the SFT phase to address the mode-averaging problem in distillation. Therefore, we assume that the main contribution on the improvements comes from changing the domain mixture in our DDK.
> >
> > We will include this discussion in our new version.

---

> > ### Author Response · Authors · 2024-08-14
> >
> > Dear **Reviewer foae**,
> >
> > Hello! We appreciate your support for the detailed suggestions on our paper. We would like to kindly ask you to take a look at our responses and reevaluate our work given our clarifications. Again, we would like to sincerely thank you very much for these constructive comments and evaluation of our manuscript.

---

### Author Rebuttal · Authors · 2024-08-06

## **General Response**

Thanks a lot for handling/reviewing our submitted manuscript. We would like to thank the reviewers for their thoughtful and constructive comments and suggestions. By addressing each of the issues raised by the reviewers, we believe that the quality and clarity of our DDK can be improved a lot. The general responses are summarized as follows:

**G.Q1: The advantages and novelties of this paper's design.**

**A1:** The pretraining datasets for  LLMs are typically sampled from a mixture of many domains. These data from different domains interplay with each other, showing complex interchangeable, unrelated, or contradictory relationships. Besides, many works have claimed that the data mixture of the pretraining data greatly affects the effectiveness of LLMs[R1,R2,R3,R4,R5,R6] and selecting the optimal domain mixture is a challenging problem. Based on knowledge distillation, our DDK aims to produce effective lightweight LLMs by transferring the domain knowledge of the teacher network. Specifically, in Fig. 1 of the main paper, we observe that the performance gap between student (Qwen-1.5 1.8B) and teacher (Qwen-1.5 14B) varies significantly across domains, which means that we need to adjust the domain mixture to better improve the weaker domains by reallocating more data sampling weights. Besides, in Table 1, Table 2, Table 7 and Table8, our DDK is better than the baseline KD a lot, which shows the effectiveness of changing the domain mixture in distillation. Therefore, we are the first to investigate the effectiveness of domain mixture for distilling LLMs and introduce the DDK to enhance the student LLMs. Meanwhile, we observe that naively changing the domain weight compromises the stability of the distillation (See Fig.3) and propose the factor smooth updating strategy to stabilize the distillation.

Overall, our DDK is motivated by the effectiveness of the domain mixture on the lightweight LLMs and we design  corresponding strategies to better  transfer the domain knowledge of the teacher network.

**G.Q2: Discuss with Sheared LLaMA.**

**A2:** Datasets for  training LLMs are  sampled from many domains and the data mixture of the pretraining data greatly affects the effectiveness of LLMs[R1,R2,R3,R4,R5,R6]. We should emphasize that the motivation of adjusting domain mixtures is similar for Sheared LLaMA and our DDK, but the solved tasks and  technical details are different a lot.

First, for the solved tasks. the Sheared LLaMA is proposed for structure pruning, while our DDK is proposed for knowledge distillation on the LLMs. Therefore, our DDK is orthogonal to the  Sheared LLaMA and we can further improve the performance of small models pruned by Sheared LLaMA. In the following table, following the setting of Table 2, we propose to use the LLaMA2-13B to distill the Sheared-LLaMA-1.3B model, which is pruned by Sheared LLaMA based on LLaMA2-7B.

| Models                       | MMLU  | RACE  | W.G.  | Arc-E | Arc-C | Avg.  |
| ---------------------------- | ----- | ----- | ----- | ----- | ----- | ----- |
| Student (Sheared-LLaMA-1.3B) | 25.71 | 23.62 | 56.81 | 41.53 | 28.14 | 35.16 |
| +DDK                         | 28.25 | 27.43 | 58.12 | 44.92 | 31.85 | 38.14 |

Second, for the technical details, the sheared LLaMA needs to first fit a scaling function using three open-sourced LLaMA models (i.e., LLAMA2-7B/13B/70B) when pruning the LLaMA model series, where the fitting data points are very limited. In Sheared LLaMA, the authors also claim the estimated reference losses for different domains are also biased as the limited data points for estimating the scaling law parameters. Besides, in many real-world pruning scenarios,  we cannot usually obtain a series of models under the same training setting. Thus, we cannot predict the losses across different domains of the small size model when a series of models are not provided. In contrast, in DDK, we directly use the domain loss predicted by the teacher model as the reference loss without fitting the scaling function, where the teacher predicted losses are used as the accurate guidance to improve the student model. Besides, the Sheared LLaMA is proposed to enhance the training efficiency of the continued pre-training setting of the pruned model, the guidance of the teacher model is not included. In addition, our DDK also introduces the factor smooth updating strategy, which is not used in Sheared LLaMA and can make the change of the domain mixture more stable. Moreover, in the following table, we also provide the results of replacing our proposed factor smooth updating strategy with the strategy of Sheared LLaMA, and observe that DDK is better a lot, which further shows the effectiveness of our factor smooth updating strategy.

| Models                                                       | MMLU  | RACE  | W.G.  | Arc-E | Arc-C | Avg.  |
| ------------------------------------------------------------ | ----- | ----- | ----- | ----- | ----- | ----- |
| Student+DDK(Qwen-1.5-1.8B)                                   | 46.01 | 71.56 | 59.1  | 75.01 | 55.03 | 61.34 |
| Student+DDK using Sheared LLaMA updating strategy (Qwen-1.5-1.8B) | 45.03 | 70.16 | 57.09 | 73.98 | 53.18 | 59.88 |

[R1]. An empirical analysis of compute-optimal large language model training, https://arxiv.org/abs/2203.15556

[R2]. Data selection for language models via importance resampling, https://arxiv.org/abs/2302.03169

[R3]. Sheared LLaMA: Accelerating Language Model Pre-training via Structured Pruning, https://arxiv.org/abs/2310.06694

[R4]. RegMix: Data Mixture as Regression for Language Model Pre-training, https://arxiv.org/abs/2407.01492

[R5]. Data Mixing Laws: Optimizing Data Mixtures by Predicting Language Modeling Performance, https://arxiv.org/abs/2403.16952

[R6]. DoReMi: Optimizing Data Mixtures Speeds Up Language Model Pretraining, https://arxiv.org/abs/2305.10429

---

### Author Response · Authors · 2024-08-10
**Summarization on the Responses**

Thanks for handling/reviewing our submitted manuscript: "DDK: Distilling Domain Knowledge for Efficient Large Language Models". We would like to thank the reviewers for their insightful and constructive comments and suggestions. By addressing each of the issues raised by the reviewers, we believe that the quality and clarity of our DDK can be improved a lot. The major responses are summarized as follows:

(1). We have carefully discussed the advantages and novelties of DDK (See Reviewer foae.Q3, Reviewer EYh3.Q1)

(2). We have additionally compared the domain-enhanced KD methods and discussed the advantages over previous methods in detail (See Reviewer foae.Q1&Q4).

(3). We have provided experiments on another training dataset and discussed the scale property of teacher (See Reviewer foae.Q2).

(4). We have provided more clarification and discussion on the evaluation prompts (MMLU and Humeneval) and continual domain learning (See Reviewer EYh3.Q2&Q3&Q4)

(5). We have discussed the limitations of requiring data distribution and provided experiments based on the clustering method to address this limitation (See Reviewer oZHE.Q1).

(6). We have discussed more related works (e.g., Sheared LLaMA, LoRAShear, Skew-KLD) and conducted a detailed analysis of these works (See Reviewer EYh3.Q1, Reviewer oZHE.Q2&Q3).

(7). We have provided a more detailed analysis and clarification on the training costs, training data, and learning rate (See Reviewer 6kDi.Q1&Q2&Q3).

Again, we would like to sincerely thank you very much for these constructive comments and evaluation for our manuscript.

---

### Author Response · Authors · 2024-08-10
**Looking forward to feedback on the Responses**

Dear Reviewers:

Hello! Since the discussion period has started, we would like to kindly ask you to take a look at our responses and reevaluate our work based on our clarifications. Please let us know whether our response addresses your concerns or whether there is any further detail we can provide to help address them. We appreciate your time and consideration!

---

### Comment · Area_Chair_QPc8 · 2024-08-12

Dear reviewers: as you are aware, the reviewer-author discussions phase ends on Aug 13. We request you to kindly make use of the remaining time to contribute productively to these discussions. If you have not read and/or responded to author rebuttal, please do it asap so that the authors get a chance to respond to you. If you have more questions to ask or want further clarification from the authors, please feel free to do it.

---

### Decision · Program_Chairs · 2024-09-25

**Decision:**

Accept (poster)

**Comment:**

The paper works on language model distillation. The paper is a borderline submisison. All reviewers agree to accept the submission, although with borderline accept or weak accept. The distillation of LLMs is a hot and useful topic of LLM research. Although the novelity of the paper is limited, it provides a lot of empirical studies, which would be valuable for other researchers. Besides, the reviewers' comments can be included in the final version in order to improve the overall papre quality.